# Epidemiology, Diagnosis and Management of Penile Cancer: Results from the Spanish National Registry of Penile Cancer

**DOI:** 10.3390/cancers15030616

**Published:** 2023-01-19

**Authors:** Ángel Borque-Fernando, Josep Maria Gaya, Luis Mariano Esteban-Escaño, Juan Gómez-Rivas, Rodrigo García-Baquero, Fernando Agreda-Castañeda, Andrea Gallioli, Paolo Verri, Francisco Javier Ortiz-Vico, Balig Fawwaz Amir-Nicolau, Ignacio Osman-Garcia, Pedro Gil-Martínez, Miguel Arrabal-Martín, Álvaro Gómez-Ferrer Lozano, Felix Campos-Juanatey, Félix Guerrero-Ramos, Josè Rubio-Briones

**Affiliations:** 1IIS-Aragón, Miguel Servet University Hospital, Universidad de Zaragoza, 50009 Zaragoza, Spain; 2Fundació Puigvert, Autonomus University of Barcelona, 08193 Barcelona, Spain; 3Applied Mathematics Department, Escuela Universitaria Politécnica de La Almunia, Universidad de Zaragoza, 50100 La Almunia de Doña Godina, Spain; 4Hospital Universitario La Paz, 28046 Madrid, Spain; 5Hospital Universitario Puerta del Mar, 11009 Cádiz, Spain; 6Hospital Germans Trias y Pujol, 08916 Badalona, Spain; 7Principe Asturias University Hospital, 28805 Alcala de Henares, Spain; 8Nuestra Señora de la Candelaria University Hospital, 38010 Santa Cruz de Tenerife, Spain; 9Virgen del Rocio University Hospital, 41013 Sevilla, Spain; 10Granada University Hospital, 18071 Granada, Spain; 11Instituto Valenciano de Urologia (IVO), 46009 Valencia, Spain; 12Marques de Valdecilla University Hospital, 39008 Santander, Spain; 13Hospital Universitario 12 de Octubre, 28041 Madrid, Spain; 14Clinica de Urología, Hospital VITHAS 9 de Octubre, 46015 Valencia, Spain

**Keywords:** penile cancer, cancer, epidemiology, incidence

## Abstract

**Simple Summary:**

Penile cancer is a rare malignant tumor mainly affecting adult/older men. Given the rarity of this disease, affecting 1/100,000 men every year, risk factors and diagnostic procedures are often inadequately described. This is the first multicenter study to describe the approach to this disease in Spain. The results highlight the need for the institution of referral centers and standardized diagnostic pathways for the optimal management of this disease.

**Abstract:**

Introduction: Penile cancer (PC) is a rare malignancy with an overall incidence in Europe of 1/100,000 males/year. In Europe, few studies report the epidemiology, risk factors, clinical presentation, and treatment of PC. The aim of this study is to present an updated outlook on the aforementioned factors of PC in Spain. Materials and Methods: A multicentric, retrospective, observational epidemiological study was designed, and patients with a new diagnosis of PC in 2015 were included. Patients were anonymously identified from the Register of Specialized Care Activity of the Ministry of Health of Spain. All Spanish hospitals recruiting patients in 2015 were invited to participate in the present study. We have followed a descriptive narration of the observed data. Continuous and categorical data were reported by median (p25th–p75th range) and absolute and relative frequencies, respectively. The incidence map shows differences between Spanish regions. Results: The incidence of PC in Spain in 2015 was 2.55/100,000 males per year. A total of 586 patients were identified, and 228 patients from 61 hospitals were included in the analysis. A total of 54/61 (88.5%) centers reported ≤ 5 new cases. The patients accessed the urologist for visually-assessed penile lesions (60.5%), mainly localized in the glans (63.6%). Local hygiene, smoking habits, sexual habits, HPV exposure, and history of penile lesions were reported in 48.2%, 59.6%, 25%, 13.2%, and 69.7%. HPV-positive lesions were 18.1% (28.6% HPV-16). The majority of PC was squamous carcinoma (95.2%). PC was ≥cT2 in 45.2% (103/228) cases. At final pathology, PC was ≥pT2 in 51% of patients and ≥pN1 in 17% of cases. The most common local treatment was partial penectomy (46.9% cases). A total of 47/55 (85.5%) inguinal lymphadenectomies were open. Patients with ≥pN1 disease were treated with chemotherapy in 12/39 (40.8%) of cases. Conclusions: PC incidence is relatively high in Spain compared to other European countries. The risk factors for PC are usually misreported. The diagnosis and management of PC are suboptimal, encouraging the identification of referral centers for PC management.

## 1. Introduction

Penile cancer (PC) is a rare cancer globally, but great variation exists between countries. Its occurrence leads to strong physical and psychological consequences for the patient since surgical treatment includes the partial or total removal of the external genitalia. The overall incidence in Europe accounts for 1/100,000 males per year [1,2]. However, there is evidence that PC incidence has been increasing in recent years in Europe [3,4]. Although it can affect men of any age, it is usually diagnosed in the sixth and seventh decades [5]. Squamous cell carcinoma (SCC) comprises the vast majority of cases accounting for up to 95% of cases of PC [6], while other variants such as basaloid, warty, verrucous, papillary, and mixed carcinoma account for the residual percentage [7]. The European Association of Urology (EAU) guidelines for PC recommend treating the patient with a partial or total penectomy and a strict follow-up depending on TNM staging and grading [7]. Multiple risk factors have been identified and described, such as smoking, lack of circumcision, phimosis, obesity, poor hygiene, and lichen sclerosis [8]. The exposure to Human Papilloma Virus (HPV) is directly related to increased reporting, with HPV DNA prevalence in 50.8% of PC patients, as reported by Saraiya et al. in a meta-analysis based on 52 studies [9]. Methodological drawbacks have been proposed in the evaluation of the association between HPV and PC due to a limited number of reliable studies, different and/or unreliable detection methods, and confusing bias due to the inclusion of any anogenital warts as premalignant lesions. However, the pooled relative risks indicate up to a 4.5-fold increased risk between seropositivity for HPV infection and invasive PC [10]. The difference in HPV prevalence among geographical regions and the low socio-economic status linked to a lack of education and seeking medical treatment might influence the profile of patients with PC [11]. In fact, the incidence rates vary dramatically among different populations, with less developed nations having the highest incidence. Reports indicate the largest disease burden concentration in Brazil, with a recorded incidence of 2.8–6.8 per 100,000 and an incidence of 0.3–0.6 per 100,000 in the United States and the United Kingdom [12,13].

In Europe, few studies reported the epidemiology of PC in different countries, analyzing the incidence trends without focusing on risk factors, clinical presentation, and treatment of this rare disease. This lack of evidence translates into a disregard for the direct impact of PC in developed countries. 

Therefore, the aim of this study is to present an updated transversal outlook on the incidence, distribution per region, risk factor profile, and clinical management of PC in Spain.

## 2. Material and Methods

### 2.1. Patient Recruitment

A multicentric, retrospective, observational epidemiological study was designed. Booth’s request for data extraction from the Register of Specialized Care Activity, including all patients diagnosed with PC in 2014 and 2015, was sent to the Ministry of Health, Social Services, and Equality of Spain. The International Classification of Diseases, Ninth Revision, Clinical Modification (ICD-9-CM) diagnostic codes were used to select potential candidates for the study; in particular, the codes 187.1 “Malignant Neoplasm of Prepuce”, 187.2 “Malignant neoplasm of glans penis”, 187.3 “Malignant neoplasm of body of penis”, 187.4 “Malignant neoplasm of penis, unspecified”, 222.1 “Benign neoplasm of penis”, 233.5 “Carcinoma in situ of penis” were used. 

The inclusion criteria were patients with a new diagnosis of PC in 2015 with hospital admission. According to the specific consultation carried out in the Health Activity Information and Statistics Area of the Institute of Health Information (Ministry of Health, Social Services, and Equality), 943 potential patients with a diagnosis of PC and hospital admission in 2015 were identified, and 753 cases in 2014. After eliminating hospital readmission in 2015 of the same patient or previous admission and diagnosis in 2014, 586 patients were identified and were considered the potential target population of our study. From the Ministry of Health database, those patients could be located anonymously. The corresponding hospitals from the patients’ locations were invited by the investigators to fill out the electronic Case Report Form for each patient (Plataforma de Investigación de Estudios Multicéntricos de la Asociación Española de Urología, PIEM-AEU. https://piem.aeu.es/proyectos/RegNCaPene/ (accessed on 2 January 2019) through a pseudonymization procedure (Appendix A). All 226 Spanish public and private centers were eligible to participate. The study protocol was approved by the Ethical Committee of Fundación Instituto Valenciano de Urología (protocol number: AEU/PIEM/2014/0002; date of approval, 27 September 2017). The study was conducted in accordance with the Declaration of Helsinki (1964) and the Guidelines for Best Clinical Practice.

Patients with a clinical diagnosis of PC in 2015 without any histology were excluded from the study. Premalignant lesions treated on an outpatient basis were not taken into account as there were no reliable documented hospital admissions, and every hospital recorded them in a heterogeneous manner.

### 2.2. Objective

The primary endpoint of the study was to evaluate the incidence of PC in a contemporary national series. 

The secondary endpoint was to describe the characteristics of the diagnosis of PC in Spain and to analyze therapeutic management and deviations from clinical practice guidelines.

### 2.3. Statistical Analysis

The PC incidence rate was reported as cases per 100,000 persons. An incidence map shows differences between Spanish regions.

The analyzed variables included demographic data, risk factors (phimosis, HPV, smoking status), sexual habits, primary lesion and lymph node status at first diagnosis, radiological and nuclear imaging, penile and sentinel lymph node biopsy histology, pathological stage according to TNM classification, and surgical treatment and complications. Pathological data were recruited following EAU recommendations. There was no central pathological review.

The continuous variables were described using median and interquartile range (p25th–p75th), and categorical variables were reported by absolute and relative frequency.

## 3. Results

The overall incidence of PC in Spain in 2015 was 2.55/100,000 male inhabitants per year. By autonomous regions, the higher incidence was observed in Castilla León, followed by Castilla La Mancha and Galicia. All three together were above the national average (Figure 1). The lowest incidence was in Cantabria, the Balearic Islands, and Extremadura. The median age at diagnosis was 72 years (interquartile range: 62–81 y.o.).

After an invitation to review the medical charts of the 586 patients identified, 61 centers answered, and 280 patients were properly registered in the database. After a critical revision of the patients entered into the database, 52 patients were excluded for protocol violations (namely, the first PC histologic diagnosis outside 2015). Only 2/61 (3.3%) hospitals reported more than 10 new patients, while 54/61 (88.5%) centers reported fewer than or equal to 5 new cases in a year. The patients were sent to urologic consultation from another specialist in 45.6% (104/228) cases and from the general practitioner in 32.9% (75/228). 

### 3.1. Clinics

The patients accessed the urologist for visually assessed penile lesions (60.5%), suppuration (11.8%), pain (5.7%), and itching (6.1%) (Table 1). The lesions were localized at the glans (63.6%), balanic groove (14%), preputial skin (9.7%), preputial mucosa (7.5%), penile skin (2.6%), and urethra (2.2%) (Figure 2). 

Phimosis was present in 45.2% (103/228) of patients, while 30.7% (70/228) had been submitted to circumcision at a median age of 60 (35–74) years old. Local hygiene, smoking habits, sexual habits, HPV exposure, and history of penile lesions were reported in only 48.2%, 59.6%, 25%, 13.2%, and 69.7% of cases, respectively. Among patients who underwent risk factors assessment, 43.5% had poor local hygiene, 46.3% were smokers, 17.5% had promiscuous sexual habits, 7.1% reported HPV exposure, and 39% had a history of penile lesions. The tumors were known as HPV-positive in 28/155 (18.1%), 8/28 (28.6%) cases were HPV 16, 1 case of HPV 12, 24, 42, 51, 52, 53, 68, 70, or 82, and 11/28 (39.3%) cases were HPV subtype unknown.

### 3.2. Histology and Staging 

Table 2 depicts the clinical and pathological features of PC. The vast majority of PC were SCC (95.2%), with only one case of melanoma (0.4%), one case of undetermined carcinoma (0.4%), and 9 cases of unknown nature (4%). The most frequent SqCCs were common (62.7%), verrucous (12.3%), condylomatous (i.e., warty) (4%), and basaloid (2.2%) variants. Less frequent variants were mixed (4 cases, 1.7%), papillary (2 cases, 0.9%), adenosquamous (2 cases, 0.9%), and sarcomatoid (1 case, 0.4%) carcinomas. In 34 cases (14.9%), no specific SqCC subtype was identified. At first biopsy, the grade of the lesion was not reported or Gx in 34% (78/228) patients, while there was a similar distribution between G1 and G2 lesions, diagnosed in 60 (26.3%) and 57 (25%) patients, respectively. All the aforementioned results were confirmed at final pathology, with the tumor grading being undetermined in 25% (57/228) of cases.

Penile cancer presented as ≥cT2 in 45.2% (103/228) of cases, with the T2 stage involving more frequently the corpus spongiosum. At final pathology, the lesions were ≥pT2 in 51% (117/228) of patients with a high incidence of pT1a (54/228; 23.7%) tumors and 10.1% (23/228) pTis tumors. The patients had clinically positive inguinal lymph nodes in 18% (41/228) of cases. Out of 130 lymph node biopsies/lymphadenectomies, 39 (30%) cases were ≥pN1. A total of 12 (5%) patients had metastatic disease. The majority were lung (41.7%; 5/12) and cutaneous (16.7%; 2/12) metastasis.

### 3.3. Management

The surgery for primary treatment was partial penectomy, wedge excision, glansectomy, and total penectomy in 46.9%, 21.1%, 14.9%, and 11.4% of cases, respectively. Nonetheless, in 27/228 (11.8%) cases, the surgical margin was positive. In fact, 21.7%, 11.8%, 4.7%, and 26.9% margins were positive in wedge excision, glansectomy, partial penectomy, and total penectomy, respectively. Topic therapy with 5-fluoracile/imiquimod was rarely used (5.3%), as for laser treatment with either CO_2_ or Nd:YAG (1.7%). 

A total of 55 inguinal lymphadenectomy procedures were performed (39 bilateral and 16 unilateral). Out of these, 8 (14.5%) were laparoscopic, and the remnant (85.5%) were open, with a median of excised lymph nodes of 12 (5–16) and positive nodes of 3 (IQR: 2–6). Pelvic lymphadenectomy was performed in 13/55 (23.6%) cases, with a median of 2 (IQR: 1–4) and 7 (IQR: 3–10) positive and excised lymph nodes. 

Patients with ≥pN1 disease were treated with neoadjuvant and adjuvant chemotherapy in 3/39 (7.7%) and 9/39 (23.1%) cases, respectively. Similarly, neoadjuvant and adjuvant radiotherapy was performed in 2/39 (5.1%) and 7/39 (17.9%) patients, respectively.

## 4. Discussion

In the present study, the epidemiology and clinical management of PC was assessed in a European country. We found that the incidence of PC was 2.55/100,000 male inhabitants per year. The incidence varied significantly among different Spanish communities. Therefore, the incidence of PC in Spain is rare but relatively high compared to other countries of the European Union, as already reported in the literature [14,15]. Geographical disparities in the incidence of PC are known, with high rates concentrated in the developing world (2.8–6.8/100,000), where neonatal circumcision is low and socio-economic conditions predispose patients to multiple risk factors. The lowest incidence, as low as 0.3/100,000, has been described in Western countries [8]. The incidence in Europe has been described between 0.45–1.7/100,000 [16], but a recent study reported an increase in incidence in the UK from 1.1 to 1.33 in 100,000 men between 1979 and 2009, probably related to the global sexual revolution and greater exposure to HPV and other sexually transmitted diseases [17].

The incidence of HPV infection may be influenced by the number of female partners throughout life and also by the age of first intercourse and history of sexually transmitted diseases [1]. The lack of information recorded on these issues is limiting and could explain the national and even regional differences found in the incidence of PC. Additionally, infection by the Human Immunodeficiency Virus (HIV) surely represents one of the factors to be considered in the development of this rare disease. In fact, patients affected by HIV have an increased risk of developing penile cancer, up to 11.1 times higher, with respect to the general population. Furthermore, there’s a faster progression from intraepithelial neoplasia to cancer (i.e., six years sooner) and increased cancer-specific mortality (i.e., four times higher). In addition, this review highlighted how coinfection by HPV is more frequent in this specific population, furtherly increasing the chance of developing penile cancer [18]. As estimated by Nunez et al. [19], up to 18% (95% CI 14.3–22.1%) of patients affected by HIV are unaware of it, regardless of being men who have sex with men (MSM) or heterosexuals. For privacy reasons, data regarding HIV status were not available for a non-negligible number of cases, so this aspect was not analyzed in this study.

Given that numerous risk factors have been identified and many of them are modifiable, knowing its incidence in our country could lead to public health campaigns focused on combating smoking trends, promoting better hygiene, and seeking a wide deployment of the vaccine against HPV [20]. 

This study represents a national registry of the diagnosis and management of PC stratified by Hospital origin in a European country and a contemporary scenario. 

In 2015, only 4 out of 17 communities reported more than 20 cases each, for a total of 61.8% (141/228) of new cases in Spain. Consequently, 88.5% of hospitals treated fewer than or equal to 5 new cases in a year. 

These results outline a worrisome national picture in the current management of PC, clearly confronting the current recommendations of EAU Guidelines [7].

Unfortunately, as observed in Table 1, the main classical risk factors mentioned in the literature for the occurrence of PC cannot be properly described. Many of them were not performed (HPV characterization and HIV status) or not reflected in the medical charts; therefore, the investigators could not translate them into the case report form. One of the suggestions we could make is that epidemiological prevention of PC should start with the proper registration and knowledge of the local incidence of risk factors in every country, mostly HPV relevance.

In the majority of the communities, the patients’ access to hospitals for suspicion of PC can be considered sporadic. Since no referral centers have been identified by the national government, the treatment of PC depends on the experience of the center where the patient is accessed. This may translate into suboptimal management of the disease, either with diagnostic or curative intent [21]. Another important aspect we have to mention is that penile cancer is initially managed by general practitioners or dermatologists, referring them to a general urologist, who normally does not refer them to any referral centers as these are not predefined. For that, patients are not usually evaluated in multidisciplinary teams. As previously mentioned, the most important risk factors for PC were not systematically explored (13.2–69.7% of cases, depending on the risk factor). The grade of the lesion was undetermined in 34% of biopsies and 25% of lesions at final pathology. Despite the high rate of partial/total penectomy (58.3% of patients) in a population characterized by 48% of <cT2 PC, the rate of positive margins at final pathology was not negligible (11.8%). The current evidence provides a mean range of positive surgical margins between 6.8% and 14.5% for glansectomy [22,23], which is in line with our study (11.8%). In a recent systematic review by Pang et al. [24], however, the positive surgical margins rate was even higher, reaching 22.6% in a study by Tang et al., furtherly underlining the need for this procedure to be executed by expert surgeons [23]. However, the rate of positive surgical margins was relevant in wedge excisions and total penectomy (> 20%). Concerning other treatment modalities (e.g., radiotherapy), the small sample did not allow any sub-analysis, although recent findings highlight the advantages of radiotherapy in organ-sparing approaches [25].

Interestingly, the role of circumcision is still an object of debate. Uncircumcised patients show a higher chance of harboring <T2 primary lesions when compared with circumcised men (54.4% vs. 34.5%, *p* < 0.001) [26], and harbor a reduced number of fungi and bacteria, with consequently altered molecular and metabolic pathways [27]. These findings may be the starting point for the demonstration of the key role of the bacterial microbiome in the development of penile cancer. This difference does not seem to impact the rate of lymph nodal involvement (i.e., N-stage), suggesting that circumcision may favor the cancerogenic process without affecting the disease’s aggressiveness [26].

Moreover, the general practitioner indicated a urologic visit for PC suspicion in only 32.9% of patients. This underlines that, in the majority of cases, the diagnostic and therapeutic care pathway of PC failed at the first step. Indeed, the patient accesses the specialist after an evaluation by another specialist or independently. This may lead to a significant delay in the correct diagnosis and potentially impact the treatment choice. However, we do not know whether this issue depends on an incorrect evaluation by the general practitioner or on underestimation/misinformation of the patient. The institution of hyper-specialized referral centers and increased formation campaigns for general practitioners represent an unmet, although urgent, need [28].

Finally, the study demonstrated a low incidence of HPV-positive PC (18.1%). This is in line with the results provided by Daubisse–Marliac et al., who showed an incidence of HPV-positive PC in France of 11% of cases [29]. However, we have to recognize that HPV characterization by either PCR in fresh tissue, p16 immunohistochemistry, RNA in situ hybridization, or combinations [30] was not the rule in our country in 2015, acknowledging that it is clearly recommended by EAU Guidelines in 2022 [7]. Despite this low incidence, the pivotal role of HPV in the pathogenesis of penile cancer is indisputable and is currently the object of intense research [31].

The study is not devoid of limitations. First, it is a retrospective study based on the national cancer registry of the country. Second, no follow-up was recorded to evaluate the oncologic and functional results of PC treatment. However, the key point of the study is centered on the need to clarify the epidemiology and management of PC in a country where the disease is rare. Our results could justify the identification of referral centers for the disease within the national healthcare system. 

## 5. Conclusions

PC is a rare disease, but its incidence is relatively high in Spain compared to other European countries. 

The risk factors for PC are usually misreported. The diagnosis and management of PC are suboptimal, probably due to the sporadic treatment of the disease by many centers. These results encourage the identification of referral centers for PC management within the national healthcare system.

## Figures and Tables

**Figure 1 cancers-15-00616-f001:**
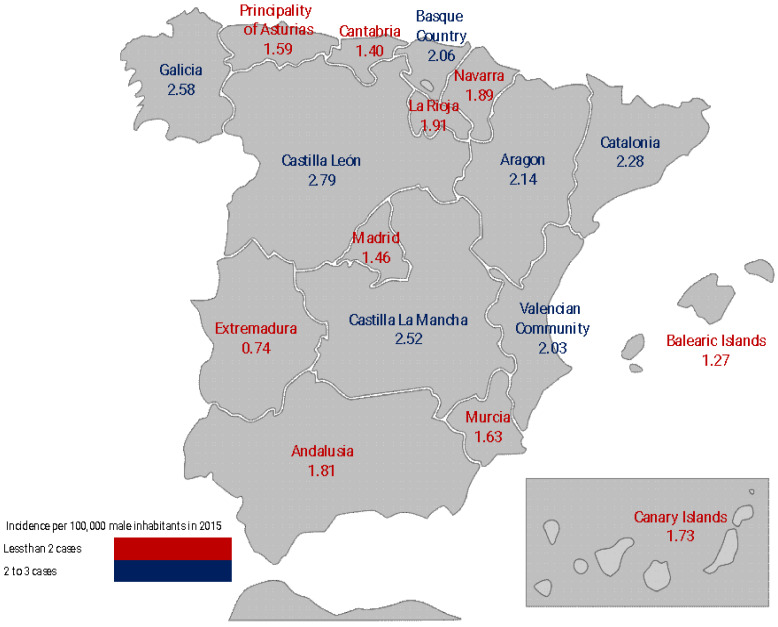
Incidence rate of penile cancer in the Autonomous Communities of Spain.

**Figure 2 cancers-15-00616-f002:**
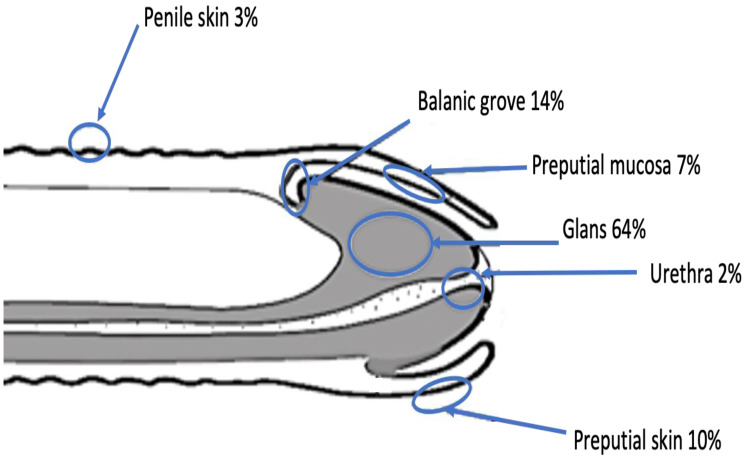
Localization of penile cancer lesions.

**Table 1 cancers-15-00616-t001:** Patients’ Baseline Characteristics (*n* = 228).

	Median (IQR) or Number (%)
Age, years	72 (62–81)
Prior circumcision	70 (30.7)
Prior HPV exposure	
Yes	2 (0.9)
No	28 (12.3)
Unknown	198 (86.8)
Local hygiene	
Correct	62 (27.2)
Poor	48 (21.1)
Unknown	188 (51.7)
Smoking habits	
Never	73 (32)
Current	63 (27.6)
Unknown	92 (40.4)
Promiscuous sexual habits	
Yes	10 (4.4)
No	47 (20.6)
Unknown	171 (75)
Prior penile lesion	
Yes	62 (27.2)
No	97 (42.5)
Unknown	69 (30.3)
Referral to a urologist	
Specialist	104 (45.6)
General practitioner	75 (32.9)
Entourage	10 (4.4)
Unknown	39 (17.1)
Symptoms	
Visually assessed penile lesion	138 (60.5)
Suppuration	27 (11.8)
Pain	13 (5.7)
Itching	14 (6.1)
Unknown	36 (15.9)
Phimosis at clinical presentation	103 (45.2)
Tumor location	
Glans	145 (63.6)
Balanic groove	17 (7.5)
Preputial skin	22 (9.7)
Preputial mucosa	32 (14)
Penile skin	6 (2.6)
Urethra	5 (2.2)
Unknown	1 (0.4)

**Table 2 cancers-15-00616-t002:** Staging and Pathological Features of Penile Cancer (*n* = 228).

	Median (IQR) or Number (%)
Histology	
Squamous cell carcinoma (SCC)	217 (95.2)
Undetermined carcinoma	1 (0.4)
Melanoma	1 (0.4)
Unknown	9 (4)
SqCC subtype	
Common carcinoma	143 (62.7)
Basaloid carcinoma	5 (2.2)
Warty carcinoma	9 (4)
Papillary carcinoma	2 (0.9)
Verrucous carcinoma	28 (12.3)
Sarcomatoid carcinoma	1 (0.4)
Mixed carcinoma	4 (1.7)
Adenosquamous carcinoma	2 (0.9)
Unknown	34 (14.9)
Clinical T stage	
cTx	23 (10.1)
cT0	5 (2.2)
cTis	22 (9.6)
cTa	7 (3.1)
cT1a	51 (22.4)
cT1b	17 (7.5)
cT2 (corpus spongiosum)	45 (19.7)
cT2 (corpus cavernosum)	21 (9.2)
cT3	28 (12.3)
cT4	9 (3.9)
Clinical N stage	
Nx	55 (24.1)
N0	132 (57.9)
N1	11 (4.8)
N2	16 (7)
N3	14 (3.1)
Clinical M stage	
Mx	35 (15.3)
M0	181 (79.4)
M1	12 (5.3)
Grade at biopsy	
Gx	78 (34.2)
G1	60 (26.3)
G2	57 (25)
G3–G4	33 (14.5)
Pathological T stage	
pTx	7 (3.1)
pT0	3 (1.3)
pTis	23 (10.1)
pTa	7 (3.1)
pT1a	54 (23.7)
pT1b	17 (7.5)
pT2 (corpus spongiosum)	50 (21.9)
pT2 (corpus cavernosum)	22 (9.6)
pT3	37 (16.2)
pT4	8 (3.5)
Pathological N stage	
pNx	98 (43)
pN0	91 (39.9)
pN1	12 (5.3)
pN2	10 (4.4)
pN3	17 (7.4)
Grade in the final specimen	
Gx	57 (25)
G1	72 (31.5)
G2	64 (28.1)
G3–G4	35 (15.4)

## Data Availability

Electronic Case Report Form for each patient Plataforma de Investigación de Estudios Multicéntricos de la Asociación Española de Urología, PIEM-AEU. https://piem.aeu.es/proyectos/RegNCaPene/ (accessed on 2 January 2019).

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
