# Peer review of "Epidemiology, Diagnosis and Management of Penile Cancer: Results from the Spanish National Registry of Penile Cancer"

_cancers, 2023, doi:10.3390/cancers15030616_

Round 1

Reviewer 1 Report

The article is very well written and the topic is very interesting despite its peculiarity. The content of the manuscript is original and of high impact in the setting of penile cancer. This is a National wide result, thus, the authors should be complimented on the work and collaboration. 

Minor revisions: 

- Correct a few English misspelling 

Author Response

We thank the reviewers for their insightful comments and suggestions regarding our manuscript “Epidemiology, Diagnosis and Management of Penile Cancer in a developed country: Results from the Spanish National Registry of Penile Cancer” which, following the suggestions of the referees, has been revised to "Epidemiology, Diagnosis and Management of Penile Cancer: Results from the Spanish National Registry of Penile Cancer”.

We greatly appreciate all the suggestions, which have been of great help in revising and improving our manuscript. We have carefully reviewed the comments and made revisions that we hope deserve your approval. We have adjusted the paper according to the comments of the reviewers.

Reviewer 1

The article is very well written and the topic is very interesting despite its peculiarity. The content of the manuscript is original and of high impact in the setting of penile cancer. This is a National wide result, thus, the authors should be complimented on the work and collaboration. 

Minor revisions: 

- Correct a few English misspelling 

Answer: We thank the reviewer for her/his comment. We have revised the manuscript to correct English misspelled words.

Reviewer 2 Report

This is an interesting descriptive study on the incidence, diagnosis, and management of rare diseases in Spain. Overall, the manuscript is written well, but the details are missing in many instances. Please see below my major and minor comments for the manuscript.

In title, I did not see the meaning of ‘developed country’ in the title. I suggest revising the title and leaving ‘developed country’ out.

Abstract

In the methods part, please remove the last sentence about ethical approval, which is not needed in the abstract. Instead, the authors must provide further detail about the total sample studied, the age range, etc.

Moreover, what were the main statistical analysis technique used and what was reported from this analysis to answer the research question should be provided briefly in one-two sentence.

Results

What was the sub-analysis and why only 228 patients were used in the sub-analysis?

Since the authors are presenting results for the year 2015, from where the per year information comes in this sentence “ 54/61 (88.5%) Centers reported ≤ 5 new cases/year”?

Overall, the results are not clear, I suggest revising it according to their research aim, first presenting the disease incidence, then association with risk factors, diagnosis/clinical presentation, and treatment in a systematic way so that readers can follow.

Conclusion

I am not sure; how did authors conclude this based on their study “The risk factors for PC are usually misreported”.

Main text

Introduction

In the first sentence, “Penile cancer (PC) is a rare malignancy in the Western countries……”, PC is also rare cancer globally, but great variation exists between countries, therefore I suggest starting the sentence from the global picture to Europe.

The abbreviated terms should be defined when they were used for the first instance, there are many such terms in the text which have not been defined.

Most of the references used in a citation are old, there are only a few references that are most recent from the last 3-4 years. I am sure there is enough new literature available on this topic globally including in European countries. Therefore, I recommend authors provide updated literature in the introduction.

The study aim should be clarified further, epidemiology is a collective term, but what authors have studied as the epidemiology of the disease should be stated in the study aim. Also, if the results/incidence are presented by regions, then that should also be mentioned.

Methods

Since this is a multicentric study, the authors should clarify how many centers/hospitals were eligible to participate in the study.

Please clarify a little more what were the diagnostic criteria/tools used to diagnose Penile Cancer in order to be included in the study.

What information on the patients provided by the hospitals should be clarified further?

The authors should describe more in detail how each of the variables used in the study (Table 1 and Table 2) were measured/defined and their source.

Currently, the statistical analysis section is missing in the methods. Please describe the statistics used to answer the research question and what was presented as a result.

Results

The results are quite descriptive, it is not clear how and on what basis the risk factors were studied.

It is not clear how the grade in the final specimen was determined. This should be clarified with the criteria used in order to understand the results.

Discussion

The discussion is unbalanced, mostly focused to discuss on the management side, but the incidence and risk factors and diagnosis as mentioned in the study aim are not well discussed.

Author Response

We thank the reviewers for their insightful comments and suggestions regarding our manuscript “Epidemiology, Diagnosis and Management of Penile Cancer in a developed country: Results from the Spanish National Registry of Penile Cancer” which, following the suggestions of the referees, has been revised to "Epidemiology, Diagnosis and Management of Penile Cancer: Results from the Spanish National Registry of Penile Cancer”.

We greatly appreciate all the suggestions, which have been of great help in revising and improving our manuscript. We have carefully reviewed the comments and made revisions that we hope deserve your approval. We have adjusted the paper according to the comments of the reviewers.

Reviewer 2

This is an interesting descriptive study on the incidence, diagnosis, and management of rare diseases in Spain. Overall, the manuscript is written well, but the details are missing in many instances. Please see below my major and minor comments for the manuscript.

In title, I did not see the meaning of ‘developed country’ in the title. I suggest revising the title and leaving ‘developed country’ out.

Answer: Following your suggestions we have removed “developed country” in the title.

Abstract

In the methods part, please remove the last sentence about ethical approval, which is not needed in the abstract. Instead, the authors must provide further detail about the total sample studied, the age range, etc.

Moreover, what were the main statistical analysis technique used and what was reported from this analysis to answer the research question should be provided briefly in one-two sentence.

Answer: We have deleted the reference to the Ethical approval. We have added the information ascertaining the descriptive nature of the paper. 

Results

What was the sub-analysis and why only 228 patients were used in the sub-analysis?

Answer: We apologize for this mistake, 228 was the sample size included in the analysis, we have corrected it in the manuscript.

Since the authors are presenting results for the year 2015, from where the per year information comes in this sentence “ 54/61 (88.5%) Centers reported ≤ 5 new cases/year”?

Answer: We thank the reviewer for her/his comment, we have replaced cases/year by cases.

Overall, the results are not clear, I suggest revising it according to their research aim, first presenting the disease incidence, then association with risk factors, diagnosis/clinical presentation, and treatment in a systematic way so that readers can follow.

Conclusion

I am not sure; how did authors conclude this based on their study “The risk factors for PC are usually misreported”.

Answer: This is an induced conclusion from the analysis of reported data in the questions/fields where risk factors were asked to the investigators

Main text

Introduction

In the first sentence, “Penile cancer (PC) is a rare malignancy in the Western countries……”, PC is also rare cancer globally, but great variation exists between countries, therefore I suggest starting the sentence from the global picture to Europe.

Answer: Following your suggestions this initial paragraph has been rewritten

The abbreviated terms should be defined when they were used for the first instance, there are many such terms in the text which have not been defined.

Answer: Following your suggestions the abbreviated terms have been defined in their first instance.

Most of the references used in a citation are old, there are only a few references that are most recent from the last 3-4 years. I am sure there is enough new literature available on this topic globally including in European countries. Therefore, I recommend authors provide updated literature in the introduction.

Answer: We have conducted an exhaustive search on this topic following the reviewer's recommendations. Most of the references identified in this new search were already included in the manuscript. Three new publications have been identified and included:

  1. Christodoulidou M, Sahdev V, Houssein S, et al. Epidemiology of penile cancer. Curr Probl Cancer 2015;39:126-36.
  2. Arya M, Li R, Pegler K, et al. Long-term trends in incidence, survival and mortality of primary penile cancer in England. Cancer Causes Control 2013;24:2169-76.
  3. Douglawi et al ,Transl Androl Urol 2017;6(5):785-790.

The study aim should be clarified further, epidemiology is a collective term, but what authors have studied as the epidemiology of the disease should be stated in the study aim. Also, if the results/incidence are presented by regions, then that should also be mentioned.

Answer: We have reshaped the last paragraph in the introduction extending the word epidemiology in a wider concept

Methods

Since this is a multicentric study, the authors should clarify how many centers/hospitals were eligible to participate in the study.

Answer: we have included the total of centers/hospital eligible to participate in the study. In fact, 226 centers.

Please clarify a little more what were the diagnostic criteria/tools used to diagnose Penile Cancer in order to be included in the study.

Answer: These criteria/tools have been rearranged and rewritten to make them easier to understand.

What information on the patients provided by the hospitals should be clarified further?

Answer: We have included as Supplementary Material: Table S1, Template of the electronic Case Report Form

The authors should describe more in detail how each of the variables used in the study (Table 1 and Table 2) were measured/defined and their source.

Answer: The Table S1 and a more detailed description in ‘Statistical analysis’ has been included to clarify this aspect.

Currently, the statistical analysis section is missing in the methods. Please describe the statistics used to answer the research question and what was presented as a result.

Answer: To clarify statistics and answer the research question, we have added two sentences:

  • The PC incidence rate was reported by cases per 100.000 persons. An incidence map shows the difference between Spanish regions.
  • The continuous variables were described using median and interquartile range (p25th-p75th), categorical variables were reported by absolute and relative frequency.

Results

The results are quite descriptive, it is not clear how and on what basis the risk factors were studied.

Answer: We present a retrospective, descriptive and cross-sectional study of anonymous epidemiological data obtained at the national level. Its descriptive nature does not allow for statistical inference or analysis of association or causality that selects risk factors from our study.

Recognized risk factors were detailed extensively in the case report form (Table S1), but could not be completed by investigators if they had not been performed or analyzed when the clinical case occurred. This is a limitation of its retrospective nature, and the loss of standardization in the collection of clinical data on penile cancer.

It is not clear how the grade in the final specimen was determined. This should be clarified with the criteria used in order to understand the results.

Answer: We included the following clarifying sentence:

  • Pathological data were recruited following EAU recommendations. There was no central pathological review.

Discussion

The discussion is unbalanced, mostly focused to discuss on the management side, but the incidence and risk factors and diagnosis as mentioned in the study aim are not well discussed.

Answer: The following phrase has been added following your suggestion:

  • Unfortunately, as observed in table 1, main classical risk factors rationed in the literature to the occurrence of PC cannot be properly described. Many of them were not performed (HPV characterization) or not reflected in the medical charts, so the investigators couldn´t translate them to the case report form. One of the suggestions we could drive is that epidemiological prevention of PC should start on the proper registration and knowledge of local incidence of risk factors in every country, mostly HPV relevance

Reviewer 3 Report

Dear Authors,

I would like to congratulate you on very interesting work.  In my assessment the Discussion part should be expanded in following:

- commentary why penile cancer is more common in the Spanish population

- reference to other publications in the aspect of suboptimal treatment of penile cancer? Are similar problems also occurring in other countries?

- I would like to receive some more information if decision about oncological treatment in Spain is taken by one person independently, e.g. by a urologist or multidisciplinary.

- I would also like to know whether the centres of reference, mentioned by the authors, are academic units and if they are in some particular region of Spain.

Having implemented complementary information, I recommend work to the further editing process. Regards.

Author Response

We thank the reviewers for their insightful comments and suggestions regarding our manuscript “Epidemiology, Diagnosis and Management of Penile Cancer in a developed country: Results from the Spanish National Registry of Penile Cancer” which, following the suggestions of the referees, has been revised to "Epidemiology, Diagnosis and Management of Penile Cancer: Results from the Spanish National Registry of Penile Cancer”.

We greatly appreciate all the suggestions, which have been of great help in revising and improving our manuscript. We have carefully reviewed the comments and made revisions that we hope deserve your approval. We have adjusted the paper according to the comments of the reviewers.

Reviewer 3

Dear Authors,

I would like to congratulate you on very interesting work.  In my assessment the Discussion part should be expanded in following:

- commentary why penile cancer is more common in the Spanish population

Answer: We add some comments in this context:

  • The incidence of HPV infection may be influenced by the number of female partners throughout life and also by the age of first intercourse and history of sexually transmitted diseases. The lack of information recorded on these issues is limiting and could explain the national and even regional differences found in the incidence of PC.
  • Given that numerous risk factors have been identified and many of them are modifiable, knowing its incidence in our country could lead to public health campaigns focused on combating smoking trends, promoting better hygiene and seeking a wide deployment of the vaccine against HPV

- reference to other publications in the aspect of suboptimal treatment of penile cancer? Are similar problems also occurring in other countries?

Answer: We have not identified any analysis in the literature in this regard. It is our hypothesis that the lack of reference centers for this infrequent pathology adds heterogeneity and with it worse results in its management, as is a recognized fact in testicular cancer.

- I would like to receive some more information if decision about oncological treatment in Spain is taken by one person independently, e.g. by a urologist or multidisciplinary.

Answer: We add the following comment on your suggestion in the discussion:

  • Another important aspect we have to mention is that penile cancer is initially managed by general practitioners or dermatologists, referring them to any general urologist, who normally do not refer them to any referral centers as these are not predefined. For that, patients are not usually evaluated in multidisciplinary teams.

- I would also like to know whether the centers of reference, mentioned by the authors, are academic units and if they are in some particular region of Spain.

Answer: Unfortunately, there are no referral centers as we try to describe in the following sentence:

  • Since no referral centers have been identified by the national government, the treatment of PC depends on the experience of the center where the patient is accessed.

We postulate the advisability of identifying these referral centers for optimal management of this rare pathology.

Having implemented complementary information, I recommend work to the further editing process. Regards.

We have tried to do it carefully. Thanks

Round 2

Reviewer 2 Report

Thank you for the revised manuscript which is now much improved. My previous comments are sufficiently addressed in the revision, and I do not have any further comments to add.